# Identification of an RNA Silencing Suppressor Encoded by a Symptomless Fungal Hypovirus, Cryphonectria Hypovirus 4

**DOI:** 10.3390/biology10020100

**Published:** 2021-01-31

**Authors:** Annisa Aulia, Kiwamu Hyodo, Sakae Hisano, Hideki Kondo, Bradley I. Hillman, Nobuhiro Suzuki

**Affiliations:** 1Institute of Plant Science and Resources (IPSR), Okayama University, Kurashiki 710-0046, Japan; annisa.aulia29@gmail.com (A.A.); khyodo@okayama-u.ac.jp (K.H.); shisano@okayama-u.ac.jp (S.H.); hkondo@okayama-u.ac.jp (H.K.); 2Graduate School of Environmental and Life Science, Okayama University, Okayama 700-8530, Japan; 3Plant Biology and Pathology, Rutgers University, New Brunswick, NJ 08901, USA; bradley.hillman@rutgers.edu

**Keywords:** mycovirus, reovirus, hypovirus, *Cryphonectria parasitica*, co-infection, RNA silencing, RNAi suppressor, chestnut blight fungus, Dicer

## Abstract

**Simple Summary:**

Host antiviral defense/viral counter-defense is an interesting topic in modern virology. RNA silencing is the primary antiviral mechanism in insects, plants, and fungi, while viruses encode and utilize RNA silencing suppressors against the host defense. Hypoviruses are positive-sense single-stranded RNA viruses with phylogenetic affinity to the picorna-like supergroup, including animal poliovirus and plant potyvirus. The prototype hypovirus Cryphonectria hypovirus 1, CHV1, is one of the best-studied fungal viruses. It is known to induce hypovirulence in the chestnut blight fungus, *Cryphonectria parasitica*, and encode an RNA silencing suppressor. CHV4 is another hypovirus asymptomatically that infects the same host fungus. This study shows that the N-terminal portion of the CHV4 polyprotein, termed p24, is a protease that autocatalytically cleaves itself from the rest of the viral polyprotein, and functions as an antiviral RNA silencing suppressor.

**Abstract:**

Previously, we have reported the ability of a symptomless hypovirus Cryphonectria hypovirus 4 (CHV4) of the chestnut blight fungus to facilitate stable infection by a co-infecting mycoreovirus 2 (MyRV2)—likely through the inhibitory effect of CHV4 on RNA silencing (Aulia et al., Virology, 2019). In this study, the N-terminal portion of the CHV4 polyprotein, termed p24, is identified as an autocatalytic protease capable of suppressing host antiviral RNA silencing. Using a bacterial expression system, CHV4 p24 is shown to cleave autocatalytically at the di-glycine peptide (Gly214-Gly215) of the polyprotein through its protease activity. Transgenic expression of CHV4 p24 in *Cryphonectria parasitica* suppresses the induction of one of the key genes of the antiviral RNA silencing, dicer-like 2, and stabilizes the infection of RNA silencing-susceptible virus MyRV2. This study shows functional similarity between CHV4 p24 and its homolog p29, encoded by the symptomatic prototype hypovirus CHV1.

## 1. Introduction

Although RNA silencing is increasingly believed to be the primary host defense against viruses in fungi, it has only been shown in a few fungi (as exemplified in the model filamentous ascomycetous fungi, *Cryphonectria parasitica* [1,2], *Fusarium graminearum* [3,4], *Magnaporthe oryzae* [5], *Colletotrichum higginsianum* [6], *Neurospora crassa* [7], and *Sclerotinia sclerotiorum* [8,9,10]). There are interesting similarities and dissimilarities in antiviral RNA silencing among these fungi. The key enzyme genes associated with fungal, antiviral RNA silencing are generally transcriptionally upregulated by virus infection [2,4,11], while their induction magnitude is different between host fungi [7,12]. Specific fungal genes that are required for antiviral RNA silencing vary depending on fungal species. In *C. parasitica* (chestnut blight fungus), only one *dicer-like* (*dcl2*) and one *argonaute-like* (*agl2*) gene play central roles [1,2,13], and no *RNA-dependent RNA polymerase* (*rdr*) genes are required for antiviral RNA silencing [14], unlike in plants [15,16,17]. In other fungi, additional *dicer* and *agl* genes also contribute to antiviral RNA silencing [3,7,8].

Plant viruses encode RNA silencing suppressors, as exemplified by potyvirus HC-Pro, tombusvirus P19, and luteovirus P0, which bind RNA silencing key players to impair their functional roles or to lead their proteolytic degradation [18,19,20]. As in the case for plant viruses, some fungal viruses have been shown to encode RNA silencing suppressors. The best-studied example is a multifunctional protein, p29, encoded by the hypovirus CHV1 (Cryphonectria hypovirus 1, the prototype virus of the family *Hypoviridae*) that confers hypovirulence to the host fungus. The functional roles attributed to this protein include roles as a papain-like cysteine protease [21,22], symptom determinant [23,24], part of the internal ribosome entry site (IRES) [25], as well as an RNA silencing suppressor [26]. CHV1 p29 suppresses RNA silencing by reducing the transcriptional induction level of *dcl2* and *agl2* upon virus infection. A few other RNA silencing suppressors have previously been identified from fungal viruses, including the S10 gene of a multi-segmented dsRNA virus, Rosellinia necatrix mycoreovirus 3 (MyRV3, the family *Reoviridae*) [27], and the ORF2 gene of a capsidless ssRNA virus, Fusarium graminearum virus 1 (FgV1, the proposed family “*Fusariviridae*”) [4]. Similar to CHV1 p29, the FgV1 ORF2 suppressor with DNA binding ability appears to suppress transcriptional upregulation of the key enzyme genes, i.e., FgDICER2 (a *dcl*) and FgAGO1 (an *agl*), of *F. graminearum*. Although the MyRV3 S10-coded suppressor activity has been demonstrated *in planta* using an agroinfiltration assay, no information about how the S10 (VP10) suppressor functions in fungal cells is available.

In a previous study, we described an interesting virus/virus interplay in *C. parasitica*, in which a symptomless hypovirus, CHV4-C18 (Cryphonectria hypovirus 4, strain C-18), a member of the family *Hypoviridae*, facilitates the stable infection of a mycoreovirus, MyRV2 (mycoreovirus 2), which is susceptible to antiviral RNA silencing, and enhances its replication and its vertical transmission through asexual spores [28]. This synergistic effect of CHV4 onto MyRV2 appears to be imposed via suppression of antiviral RNA silencing by CHV4. In this current study, we show that the N-terminal portion of the CHV4-C18 polyprotein, termed p24, is a protease that autocatalytically cleaves itself from the rest of the viral polyprotein, and functions as an antiviral RNA silencing suppressor.

## 2. Materials and Methods

### 2.1. Viral and Fungal Strain

Viral and fungal strains used in this study are listed in Table 1. A field-collected strain (C18) *C. parasitica* doubly infected by CHV4-C18 and MyRV2 and derivative isogenic isolates that were either virus-free or singly infected with CHV4-C18 or MyRV2 were described [28,29,30] earlier. The standard strain EP155 of *C. parasitica* [31], and its RNA silencing-deficient mutant, *dcl2* [1] were a generous gift from Donald L. Nuss, Institute for Bioscience and Biotechnology, University of Maryland. A wild type CHV1 isolate (CHV1-EP713) and its mutant, lacking the viral silencing suppressor, p29 (CHV1-Δp69), were described previously [32,33]. Fungal strains were cultured as described earlier on potato dextrose agar (PDA) plates on the bench top at 22–26 °C for maintenance and colony phenotypic observations, and on PDA plates layered with cellophane for RNA preparation.

### 2.2. Fungal Transformation

Spheroplasts were prepared from *C. parasitica* strain C18 as described previously [34], and transformed with an enhanced green fluorescent protein (eGFP) reporter (pCPXHY-C18-*dcl2pro*::*egfp*) or CHV4 p24 expressing (pCPX-CHV4-p24) constructs. The eGFP reporter strain containing an *egfp* gene under the control of the *dcl2* promoter (~2 kb) of strain C18, termed C18 *dcl2pro*::*egfp,* was constructed as described earlier [35]. pCPX-CHV4-p24 was prepared by inserting the CHV4 p24 coding domain (amino acids 1–214) region flanked with a termination codon into the pCPXHY3 expression vector. Transformants were also prepared with the empty pCPXHY3 vector and used as a control.

### 2.3. Functional Expression in Escherichia coli of the Putative p24 Proteinase Domain of CHV4

The N-terminal region of the CHV4 polyprotein (map positions 287 to 1706; encoding amino acids 1 to 473, which include a coding region of the tentative protease p24, amino acid position 1 to 214, with the estimated molecular weight of 52 kDa) was amplified by RT-PCR and cloned into the expression vector pCold I (Takara Bio, Kusatsu, Japan) that had been modified to include GST (glutathione S transferase) in the expression cassette. The CHV4-derived 52kDa coding region was fused in frame with a His (hexa histidine)-tag with GST at the N-terminus and an HA (hemagglutinin)-tag at the C-terminus (pCold-His/GST-CHV4 52kDa-HA). *Escherichia coli* strain BL21 (DE3) (Promega, Madison, WI, USA) was transformed with this construct. Induction of the recombinant protein with IPTG (isopropyl-β-d-thiogalactopyranoside) at 15 °C for 4 h was according to the manufacturer’s instructions (Takara Inc., Kusatsu, Japan). The affinity-purified recombinant protein was examined by SDS-polyacrylamide gel electrophoresis (PAGE).

Proteins in purified preparations separated on an SDS-PAGE gel were transferred onto polyvinylidene difluoride (PVDF) membrane using 10 mM CAPS buffer containing 10% methanol and followed with amido black staining (0.1% amido black, 10% acetic acid, 40% methanol). The specific band obtained from amido black staining was then excised from the membrane and processed for amino acid sequences on a gas-phase protein sequencer Shimadzu Model PPSQ-31A (Kyoto, Japan) at the Department of Instrumental Analysis and Cryogenics, Advanced Science Research Center, Okayama University (Okayama, Japan).

### 2.4. RNA Extraction, RT-PCR and RNA Blotting

Total RNA extraction and Northern blotting were performed as described previously [34]. Digoxigenin (DIG)-labeled DNA probes prepared by PCR (PCR DIG Labeling Mix, Roche) were used to detect targeted RNAs, i.e., *dcl2* mRNA, MyRV2-S10 segment, and CHV4-C18. The accumulation levels of targeted RNAs were quantified by ImageJ software (Version 1.52v, National Institutes of Health, Bethesda, MD, USA) and normalized to those of ribosomal RNA (28S rRNA). To confirm the virus infection, conventional RT-PCR and one-step colony RT-PCR were carried out as described earlier using QuickTaq (Toyobo, Osaka, Japan) or PrimeScript^®^ OneStep RT-PCR Ver.2 (Takara Bio), respectively [28,36]. Appendix A lists all primers used in this study. RT-qPCR was performed as described by Hyodo et al. [37] to quantify *dcl2* mRNA and MyRV2 S4 mRNA. Primer pairs, double-stranded RNAs were isolated by the method of Eusebio-Cope and Suzuki [34].

### 2.5. Small RNA Analysis

Total RNA extracted from virus-infected *C. parasitica* C18 strain was subjected to small RNA sequencing analysis. Small RNA cDNA library preparation and subsequent deep sequencing with Illumina platform (San Diego, CA, USA) (HiSeq 2500; 50-bp single-ends reads) were conducted by Macrogen Inc. (Tokyo, Japan). Raw reads (total read numbers: CHV1-EP713/C18 = 33,766,667; CHV4-C18/C18 = 36,794,504; MyRV2/C18 46 = 892,744; CHV4-C18 + MyRV2/C18 = 40,103,587) were trimmed by removing the adapter with low-quality base and size filtering (15 to 32 nt in length), the retained reads in each library were mapped into each virus genome (CHV1-EP713 and CHV4-C18) or respective genomic segments (MyRV2) using CLC Genomic Workbench (version 11; CLC Bio-Qiagen, Aarhus, Denmark). The mapped virus-derived small RNA reads were used for in-depth analysis with the program MISIS-2 [38].

### 2.6. Confocal Laser Scanning Microscopy of the Reporter Fungal Strain Infected by Diverse Viruses

A micro cover glass (22 mm × 22 mm, thickness 0.12–0.17 mm) (Matsunami, Osaka, Japan) was placed on the top of cellophane-overlaid PDA. A small plug of the fungal culture of the GFP reporter strain infecting each of the tested viruses (MyRV2, CHV4-C18, CHV1-EP713, or CHV1-∆p69) was placed at the edge of micro cover glass, and allowed to grow for three days. After the mycelia grew between the micro cover glass and cellophane, the clover glass with mycelia was detached and placed on top of a micro slide glass (Matsunami). GFP expression was observed using Olympus Fluoview FV1000 confocal laser scanning microscope (Olympus, Tokyo, Japan).

## 3. Results

### 3.1. Cryphonectria Parasitica Hypovirus 4 Suppresses Upregulation of dcl2

It was previously shown that CHV4 suppresses the transcriptional upregulation of *dcl2* induced by co-infecting MyRV2 in *C. parasitica* strain C18 [28]. Recently, an eGFP-based reporter system for monitoring *dcl2* transcript levels was developed in the genetic background of standard *C. parasitica* strain EP155, in which the *dcl2* promoter was linked to an *egfp* gene (pCPXHY-*dcl2pro*::*egfp*) [39]. The 2-kbp *dcl2* promoter region of strain C18 was used to replace the corresponding region of pCPXHY-*dcl2pro*::*egfp* (namely pCPXHY-C18-*dcl2pro::egfp*). Fungal transformants expressing this construct in the virus-free C18 genetic background were prepared and fused with C18 isolates singly infected by CHV4-C18, MyRV2 and CHV1-p69, respectively, or doubly infected by CHV4-C18 and either MyRV2 or CHV1-p69 (Table 1). First, the RNA-silencing suppressor activities of CHV4-C18 in *dcl2* transcriptional upregulation were confirmed using the reporter system. Upon infection of the reporter fungal strain by the strong RNA silencing inducer viruses, CHV1-p69 or MyRV2, the fungal strain fluoresced brightly (Figure 1A, top row). However, this green fluorescence was significantly reduced by co-infection by CHV4 (Figure 1B). Similar to CHV1 infection, no green fluorescence was detectable in the reporter fungal strain infected by CHV4. A similar eGFP fluorescence pattern was observed in an independent set of fungal strains (Appendix A). The reduction in levels of *dcl2* and *egfp* transcripts by CHV4 infection was also confirmed by northern blotting (Figure 1C), in which the transcript levels were reduced by four to seventeen-fold.

These results confirm that CHV4 is able to impair the otherwise strong transcriptional induction of *dcl2* triggered by infection with heterologous viruses (MyRV2 or CHV1-p69) that otherwise induce antiviral RNA silencing.

### 3.2. Co-infection Results in Alterations in Profiles of Virus-Derived Small RNAs

Previous results have shown that the disruption of *dcl2* in *C. parasitica* affects the small RNA profile of CHV1-∆p69 [40]. To investigate the small RNA profiles of CHV4-C18 and MyRV2 in single and double infections, small RNAs were deep-sequenced. The *C. parasitica* strain C18 infected with CHV1-EP713, a hypovirus with a well-characterized RNA silencing suppressor, was used as a reference. CHV1 derived small RNAs showed lesser accumulation in their negative-strands (27%) than the positive-strands (73%), both having a peak of 21 nt as reported by Zhang et al. [12] (Figure 2A,B).

Interestingly, single infection by CHV4-C18 showed a unique small RNA profile that has a size distribution without a sharp 21-nt peak in the positive-strand small RNAs, no preferential nucleotide at their 5′ termini, and a high preponderance of the small RNAs being of positive-strand polarity (94.5%) (Figure 2A,C, Appendix A). This is similar to the profile in the *dcl2* mutant in the EP155 genetic background when infected with either CHV1-p69 or a mitochondrially replicating mitovirus CpMV1 (Cryphonectria mitovirus 1) [40,41]. This unusual pattern of CHV4 small RNA profile may be associated with the lower accumulation of viral replicative dsRNA and no or very weak *dcl2* induction in the fungal hosts [28,42]. Co-infection by MyRV2 led to enhanced accumulation of both strands of CHV4-derived small RNAs with a clear 21-nt peak (Figure 2A,C), likely through modest induction of *dcl2* by MyRV2 compared to single infection by CHV4 (Figure 1C). An almost equal amount of small RNAs derived from the MyRV2 positive and negative strands accumulated in single or double infections (Figure 2A,C). MyRV2-derived small RNAs had a clear peak at 21-nt in both the positive and negative strands. Co-infection with MyRV2 led to CHV4-derived small RNA distribution with the size of 21 nt as the highest peak showed for both small RNA strands (Figure 2C). Co-infection by CHV4 had a modest positive impact on the accumulation of both strands of MyRV2-derived small RNAs, i.e., approximately two-fold increase observed in double infections relative to single infections (Figure 2A). This increase appeared to be due to the CHV4-mediated suppression of *dcl2* transcriptional induction by MyRV2 and increased MyRV2 accumulation (see below). The relative accumulation of virus-derived small RNAs and *dcl2* was inversely correlated, and yet was non-proportional, as reported earlier by Andika et al. [35].

### 3.3. CHV4 Encodes a Papain-Like Cysteine Protease p24

The family *Hypoviridae* contains four species whose exemplar strains were derived from *C. parasitica*: CHV1, CHV2, CHV3, and CHV4 [43]. CHV2 and CHV3 have been shown to encode one papain-like cysteine protease at the most N-terminal portion of the polyprotein, while CHV1 possesses two proteases. Linder-Basso et al. [42] characterized the prototype CHV4 isolate SR2 (CHV4-SR2) and predicted the catalytic cysteine (Cys173) and histidine residues (His223), and self-cleavage di-glycine site (Gly245 Gly246). Although the two CHV4 isolates from USA, SR2 and C18, share over 99% nucleotide sequence identity, a single nucleotide polymorphism in the 5’ proximal region results in different initiation codon positions 194 for CHV4-SR2 and positions 287 for CHV4-C18, leading a difference in polyprotein size (31 codons shorter in CHV4-C18) [28]. To examine the possible biological function of CHV4-C18 p24, we referred to available information on a well-established hypoviral papain-like cysteine protease CHV1 p29 [21,22], which also acts as an RNA silencing suppressor [26], a symptom determinant [23,24,44], and part of an IRES element [25].

To test the hypothesis that CHV4-C18 p24 is a papain-like cysteine protease, an *E. coli* expression construct was prepared, as shown in Figure 3A. The N-terminal fragment of the CHV4-C18 polyprotein (52 kDa polypeptides), including the tentative protease p24 coding domain, which carried a His-tag and GST at the N-terminus and an HA-tag at the C-terminus, was expressed in *E. coli* cells. The expected total molecular mass of the recombinant protein (His/GST-CHV4 52kDa-HA) is ~84 kDa.

To test whether CHV4 p24 had auto-proteolytic activity, total protein fractions of *E. coli* cells transformed by the pCold-construct were examined by SDS-PAGE Figure 3B. Two protein bands of 58 kDa and 25 kDa were specifically induced in transformed *E. coli* cells upon induction by cold shock and IPTG treatment. From the predicted cleavage site of p24, the two products were assumed likely to be a fusion of His-GST-CHV4 p24 (~53kDa) and CHV4-remaining polypeptides (amino acids 214–473) with HA-tag (~28kDa), respectively. These two protein bands were observed neither in cells transformed with the expression construct, pCold-His/GST-CHV4 52kDa-HA, under non-induced conditions, nor in cells transformed cells with the empty vector regardless of whether under induced or non-induced conditions. The N-terminal sequence of the 25 kDa-product was determined to be G-R-E-S-D-A-D-S-H-P, identical to the CHV4-C18 polyprotein sequence at 215 to 224 and in agreement with the prediction for the CHV4-SR2 strain by Linder-Basso et al. [42] (Figure 3).

These results show that CHV4 encodes a papain-like protease, designated as p24, which cleaves itself from the N-terminal portion of the polyprotein at a di-glycine site (ARLG GRES) likely in a co-translational manner as in the case for the peptidase C7 superfamily homologs of other *C. parasitica* hypoviruses [22,45].

### 3.4. CHV4-C18 p24 Compromises dcl2 Upregulation and Enhances MyRV2 Accumulation

The multifunctional protein CHV1 p29 was the first RNA silencing suppressor identified from mycoviruses [26,46]. CHV1 p29 exerts its suppressor activity by inhibiting the transcriptional upregulation of the RNA silencing key enzyme genes, *dcl2* and *agl*2. To investigate whether CHV4-C18 p24, a putative homolog of CHV1 p29, exhibits RNA silencing suppressor activities, the corresponding p24 ORF region was expressed in a virus-free isogenic isolate of *C. parasitica* C18 strain. Five transformants (designated as C18p24-1 to -5) were tested for the suppression of the *dcl2* transcriptional upregulation following infection by MyRV2 as an RNA silencing trigger [28]. Lower induction of the *dcl2* transcription was observed in transformants expressing CHV4-C18 p24 (29% for C18p24/MyRV2) relative to non-transformants (expressed as 100% for C18/MyRV2), and this suppression was slightly smaller than that in CHV4 co-infection (18% for C18/MyRV2+CHV4) (Figure 4, top row). Such suppression was not observed in transformants with the empty vector (Appendix A). Note that the difference in *dcl2* mRNA accumulation between the MyRV2-infected C18 wild type strain and the other fungal strains was statistically significant (Appendix A).

MyRV2 accumulation in CHV4-C18 p24 transformants, represented by the S10 segment, compared with that in non-transformant C18 in the absence or presence of CHV4-C18. Interestingly, transgenic expression of CHV4-C18 p24 enhanced MyRV2 accumulation, though not as greatly as CHV4-C18 co-infection, (Figure 4, second row). MyRV2 accumulation was slightly higher, albeit not at statistically significant levels (Appendix A), in the p24 transformants (approximately 1.6-fold relative to wild-type C18) than in non-transformants not infected by CHV4-C18, but much lower than in lines co-infected by CHV4-C18 (approximately 5-fold relative to wild-type C18) (Figure 4, second row). Enhanced MyRV2 accumulation was not observed in transformants with the empty vector pCPXHY3, which showed slightly lower MyRV2 accumulation than C18p24 (Appendix A).

### 3.5. CHV4-C18 p24 Enhances MyRV2 Stable Infection and Horizontal Transmissibility

It has previously been shown that during repeated subcultures of *C. parasitica* host strain C18, CHV4-C18 co-infection enhances the stability of MyRV2, which is otherwise susceptible to antiviral RNA silencing and is eliminated [28]. We tested whether transgenic expression of CHV4-C18 p24 alone enhanced the stability of MyRV2 using several C18p24 transformant strains (designated C18p24-1 to -5) (Figure 5). Three MyRV2 infectants for each of the five independent CHV4-C18 p24 transformants were examined, while three infectants of empty vector transformants (termed C18emp) were used as controls (Appendix A). As observed in wild-type *C. parasitica* wild-type strain C18 [28], MyRV2 was more frequently lost in the control strain as subculturing proceeded (Appendix A). By contrast, transgenic expression of CHV4-C18 p24 greatly reduced MyRV2 elimination. For example, at the 7th subculture, 11 of 15 C18p24 infectants still carried MyRV2, while only 7 of 15 control (C18emp) infectants harbored MyRV2 as determined by RT-PCR (Figure 5). By the 10th subculture, two-thirds of control (C18emp) infectants lost MyRV2, whereas 60% of p24 transformants (C18p24) retained MyRV2 (Figure 5 and Appendix A), somewhat lower than the 100% demonstrated previously for strains co-infected with CHV4-C18 [28]. MyRV2 was lost in 87% of subcultures of non-transformant C18 strains by the 10th subculture in the previous work [28], while nine out of fifteen transformants (60%) retained MyRV2 during the period of 10th subculture using similar culturing conditions (Figure 4). Thus, the degree of MyRV2 stability conferred by CHV4-C18 p24 transgenic expression was smaller than that conferred by CHV4-C18 infection, but was much greater than that in the non-transformant C18 strain.

We have noted that MyRV2 could rarely be moved laterally by anastomosis to wild type C18 in the absence of CHV4-C18. We attempted to transfer MyRV2 to virus-free C18 (C18-VF) using three plates in each of three independent co-culturing assays. We could move MyRV2 only to two recipient colonies out of nine co-cultures. To examine whether p24 alone could also facilitate the horizontal transmission of MyRV2, CHV4-C18 p24 transformants (C18p24), and non-transgenic controls were co-cultured with MyRV2-infected C18 strain (C18/MyRV2) (Figure 6). Efficient horizontal transmission of MyRV2 to recipient fungal transformants expressing p24 (C18p24) and to recipient isolates harboring the complete virus CHV4-C18, but not to virus-free wild-type strain C18 (C18-VF). A CHV4 transfer pattern similar to that observed for C18-VF was obtained. Co-culturing using five co-cultures yielded no CHV4 horizontal transfer to C18emp.

The combined results, shown in Figure 5 and Figure 6, strongly suggest that CHV4 p24 is an RNA silencing suppressor, derived from the N-terminal portion of the polyprotein encoded by CHV4.

## 4. Discussion

We have previously shown a commensal interaction between a symptomless hypovirus, CHV4-C18, and a hypovirulence-conferring mycoreovirus, MyRV2, in the *C. parasitica* strain C18, in which the former virus facilitates efficient vertical transmission and stable infection of the co-infecting second virus during subculture [28]. In the absence of CHV4-C18, MyRV2 is unstable, is lost from culture quickly, and is unable to move efficiently either vertically through conidia or horizontally by anastomosis, likely because it is highly susceptible to antiviral RNA silencing. Here, we report the identification of CHV4-C18 self-cleaving protease p24 as an RNA silencing suppressor that is directly involved in this commensalism or a one-way synergistic effect on a co-infecting virus.

As predicted for another CHV4 strain, SR2, by Linder-Basso et al. [42], the putative cleavage site of the polyprotein was mapped to the di-glycine peptide at positions 214 and 215 in CHV4-C18, and is strictly conserved among CHV4 isolates sequenced, thus far, as well as in some other hypoviruses, including CHV1, Phomopsis longicolla hypovirus 1 and Valsa ceratosperma hypovirus 1 [22,28,42,47]. A glycine-threonine dipeptide (instead of the second glycine) is the cleavage site of CHV3-GH2 p32, the homolog of CHV4-C18 p24 [45]. While this study did not explore the predicted catalytic residues of CHV4-C18 p24, cysteine 142 and histidine 192 were implicated based on their conservation with similar spacing in homologous papain-like proteases identified in many hypoviruses and related plant-infecting potyviruses [42]. It should be noted that in addition to the papain-like cysteine proteases of the peptidase C7 and C8 superfamilies in *C. parasitica* hypoviruses [48,49], putative 2A-like proteases (a self-processing peptide motif) were recently discovered in some closely related hypoviruses infecting *Rosellinia* and *Fusarium* fungi [50].

Suppression of RNA silencing is one of the strategies employed by viruses to counteract host antiviral machinery. However, identifying RNA silencing suppressors based on sequence comparison is not easy, because no common conserved sequence motifs are detectable between distant virus groups. This is particularly true for fungal viruses because no robust, versatile methods for identifying RNA silencing suppressors are available. As an antiviral defense response, key enzyme genes involved in RNA silencing (*dcl* and *agl*) are transcriptionally upregulated upon virus infection. Here, a recently developed reporter assay [39] was employed to confirm the RNA silencing suppressor activity of CHV4-C18, in which *dcl2* transcription induction was monitored by GFP fluorescence (Figure 1). The results, shown in Figure 1, further validated the method and contributed to the identification of CHV4-C18 p24 as an RNA silencing suppressor that likely potentiates the enhancement of vertical transmission and stability of an RNA silencing-susceptible virus, MyRV2. Generally, virus genomic dsRNA or viral dsRNA replicative forms can trigger antiviral RNA silencing. On the one hand, lower accumulation of the replicative dsRNA form of CHV4 [28,42] may allow the virus to evade host RNA silencing. On the other hand, CHV4 encodes RNA silencing suppressor p24 as a counter defense mechanism (Figure 4, Figure 5 and Figure 6). It is interesting to speculate whether p24 RNA silencing activity is essential for the stable CHV4 infection in the host fungus. Relevant to this question, the relatively few CHV4 variants examined to date display varying levels of virulence, and differences in RNA accumulation levels in those variants have not been examined [30,42]. Thus, it is possible that p24 silencing suppressor activity is more important for CHV4 stability in some variants than others.

Robust systems for examining RNA silencing suppression mechanisms in virus-infected plants are available, and many have been well studied [19,51]. Plant viral RNA silencing suppressors target various players or steps of RNA silencing to suppress antiviral silencing [19]. In contrast, only a limited number of RNA silencing suppressors have been identified from fungal viruses. These include CHV1 p29 [26], MyRV3 VP10 [27], and FgV1 p2 [4]. The RNA silencing suppressor CHV1 p29 was shown to suppress the transcriptional induction of RNA silencing key enzyme genes that are virus-responsive [2,52]. Similarly, the RNA silencing suppressor named p2 (the ORF2 protein) recently reported from the fungal virus FgV1, belonging to the proposed family *Fusariviridae* [9], functions via transcriptional repression mechanism like CHV1 p29. The mechanism of the MyRV3 VP10 suppressor was not determined [27]. In the case of *C. parasitica*, the upregulation of the key RNA silencing genes requires the DCL2 protein, but not AGL2, in addition to the general transcriptional co-activator, SAGA (*S*pt-*A*da-*G*cn5 *a*cetyltransferase) complex [35,40]. It remains elusive how these RNA silencing suppressors impair the transcriptional regulatory pathway in fungal cells.

While CHV1 and CHV4 are members of the family *Hypoviridae*, they show different and similar molecular and biological properties. CHV1 has two ORFs, A and B, each encoding a papain-like cysteine protease, p29 (peptidase C7 superfamily) and p48 (peptidase C8 superfamily). CHV4 has only one ORF and encodes one single protease, p24 (peptidase C7 superfamily). CHV1 has been reported from Asia, Europe, and North America and induces hypovirulence in *C. parasitica* together with associated phenotypic alterations that vary in degree depending on the fungal strains. CHV4 is generally asymptomatic and has been only reported from the USA and it is pervasive in the country [53]. Despite these differences, it is of interest to note that CHV4-C18 p24 and CHV1 p29 are both multifunctional and show interesting parallels: Both proteins can self-cleave at their C-termini, and both suppress antiviral RNA silencing via inhibition of the upregulation of the RNA silencing key genes, including *dcl2*. Additional roles assigned to CHV1 p29 include its action as a symptom determinant [23,24,44], and a part of IRES [25]. It will be an interesting future challenge to investigate whether CHV1 p29 and CHV4 p24 have the same mode of action as RNA silencing suppressors and whether another CHV1 p29 homolog, CHV3 p32, has an RNA silencing suppressor function.

## 5. Conclusions

In conclusion, we have described a functional role of CHV4 p24 in RNA silencing suppression. CHV4 p24, encoded by the 5′-terminal portion of the single large open reading frame, is cleaved from the large polyprotein by its autocatalytic protease activity between the two glycine residues at Gly214 and Gly215. Transgenic expression of CHV4 p24 led to reduced levels of the upregulation of the antiviral RNAi key genes, *dcl2*, upon infection by a trigger virus, MyRV2, and enhanced levels of MyRV2 content. Furthermore, CHV4 p24 enhanced the maintenance of MyRV2 during subculturing and horizontal transfer during co-culturing. These combined results allowed us to identify CHV4 p24 as an RNA silencing suppressor.

## Figures and Tables

**Figure 1 biology-10-00100-f001:**
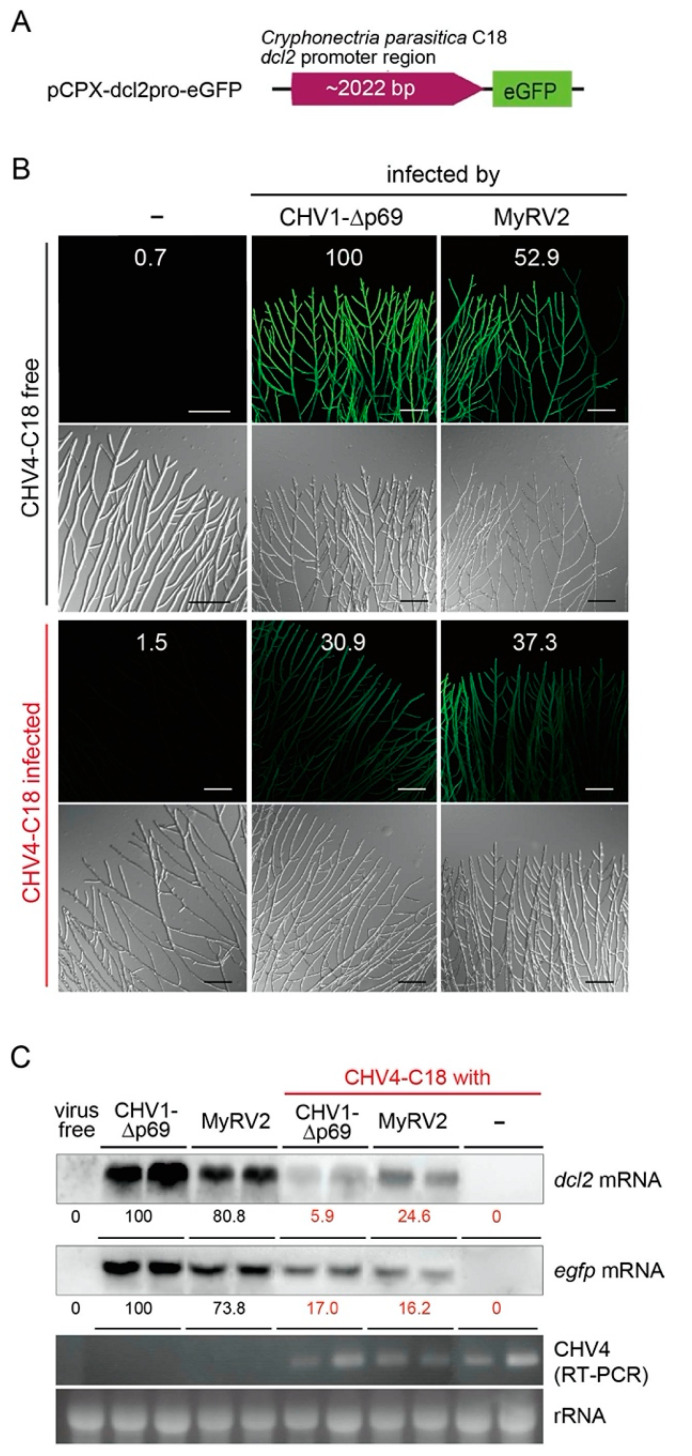
CHV4 suppresses antiviral RNA silencing via inhibiting *dcl2* transcriptional upregulation. (**A**) The organization of a reporter constructs pCPXHY-C18-*dcl2pro*::*egfp*. An *egfp* gene fused with a 2.2 kbp *C. parasitica* C18 genomic sequence containing the *dcl2* promoter region was cloned into pCPXHY1 and used to transform a virus-free isogenic isolate of *C. parasitica* C18 strain (C18-VF) [28]. (**B**,**C**) CHV4-mediated suppression of the GFP reporter induction. The reporter fungal strain with pCPXHY-C18-*dcl2pro*::*egfp* (C18/*dcl2*pro-eGFP) was infected by CHV1-∆p69 and MyRV2 (**B**), which are strong *dcl2* triggers, or CHV4 and CHV1 wild type (**C**). Values in the respective panels of (**B**) show the relative intensity of the reporter eGFP green fluorescence quantified by ImageJ, with the CHV1-∆p69-infected strain expressed as 100. Total RNA fractions were obtained from two biological replicates of the reporter fungal strain (C18/*dcl2*pro-eGFP) infected by each virus, and subjected to northern blotting to monitor *dcl2* and *egfp* transcript levels. Hybridization and probe preparation are described in the Materials and Method section. Mean values of band intensity in the northern blots (**C**) quantified by ImageJ are shown below each blot. RT-PCR detection for CHV4 infection or dsRNA electrophoretic gel analysis for MyRV2 (viral dsRNA genome) or CHV1 and CHV1-∆p69 (viral replicative dsRNA form) infection in the reporter fungal strain were conducted. Ribosomal RNA (28S rRNA) was used as a loading control.

**Figure 2 biology-10-00100-f002:**
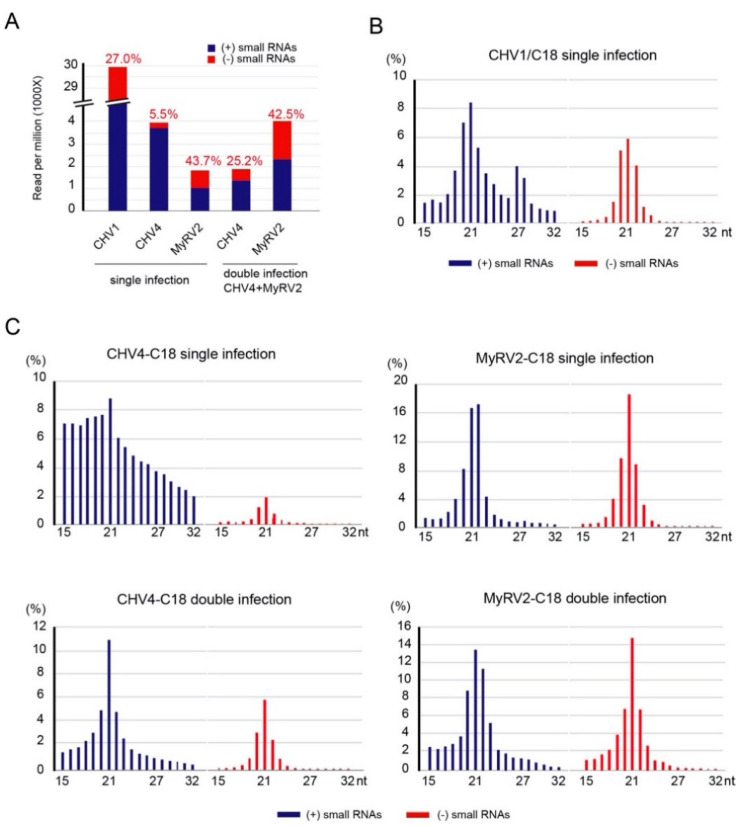
Viral-derived small RNA profiling of *C. parasitica* C18 strain in either singly or doubly infected by CHV4-C18 and MyRV2. (**A**) Sense and small antisense RNAs (15 to 32 nt) derived from singly or doubly infected CHV4-C18 and MyRV2 are shown on a per-million-total-small-RNA read (the y-axis represents the one-thousandth read numbers). The C18 strain infected singly by CHV1-EP713 (wild type) was also included for the analysis as a reference. The percentage shown on each bar represents the ratio of negative strand small RNAs. (**B**) Size distributions of small RNAs derived from CHV1 in singly infected C18. (**C**) Size distributions of virus-derived small RNAs in single and double infections of strain C18 by CHV4-C18 and/or MyRV2. Blue and red bars denote plus-strand and minus-strand small RNAs (**B**,**C**).

**Figure 3 biology-10-00100-f003:**
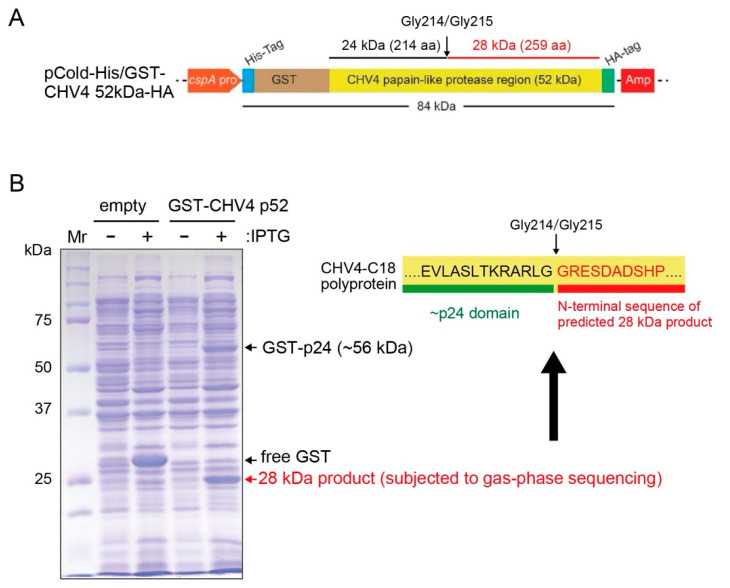
Autocatalytic proteolytic activity of CHV4 p24 encoded at the most N terminal portion of the viral polyprotein. (**A**) Schematic representation of the N terminal portion of the CHV4-C18 polyprotein. The N-terminal coding regions spanning amino acids 1 to 473 (the estimated molecular weight of ~ 52 kDa) was cloned into the *Kpn*I-*Nde*I site of an *Escherichia coli* expression vector, pCold. The 52kDa coding region was fused in frame with a His-tag and GST at the N-terminus and an HA-tag at the C-terminus (pCold-His/GST-CHV4 52kDa-HA, 86 kDa). The predicted cysteine and histidine residues and di-glycine are shown on the top. (**B**) The self-cleavage activity of CHV4-C18 p24 in the recombinant *E. coli* cells. Total proteins in *E. coli* cells transformed by the CHV4-52kDa expression construct and empty vector were electrophoresed in SDS-PAGE gel and stained with Coomassie Brilliant Blue. Two protein bands of 52kDa and 28 kDa were specifically induced in the recombinant cells with the CHV4-52kDa construct upon induction (left panel, shown with arrows). Unpurified fractions of the over-expressed protein encoded by the N terminal portion of CHV4 were blotted onto PVDF membrane, and the 28 kDa protein band was subjected to chemical sequencing. The six determined amino acid residues were the same as those deduced from the CHV4 nucleotide sequence. The cleavage site was experimentally identified as Gly245↓Gly246, which had been predicted by Linder-Basso et al. [41].

**Figure 4 biology-10-00100-f004:**
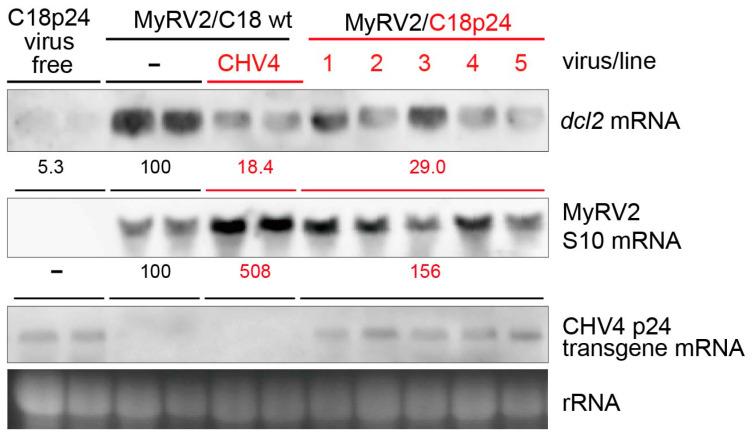
Suppression of transcriptional upregulation of *dcl2* by CHV4-C18 p24. Total RNA fractions were obtained from fungal strains shown on the top. *C. parasitica* C18 non-transformants infected by MyRV2 (C18/MyRV2), and MyRV2 plus CHV4 (C18/MyRV2+CHV4), CHV4 p24 transformants uninfected (C18p24-VF) or infected by MyRV2 (C18p24/MyRV2). Four independent transformants (1 to 5) were tested in this experiment. Total RNA was examined by northern blotting for *dcl2* transcripts, MyRV2 S10 mRNA, and CHV4-C18 p24 transgene transcripts. Ribosome RNA (rRNA) serves as a loading control. Mean values of band intensity in the northern blots quantified by ImageJ are shown below each blot. Note that the presence of CHV4 genomic RNA was confirmed in fungal strain C18 doubly infected MyRV2 and CHV4.

**Figure 5 biology-10-00100-f005:**
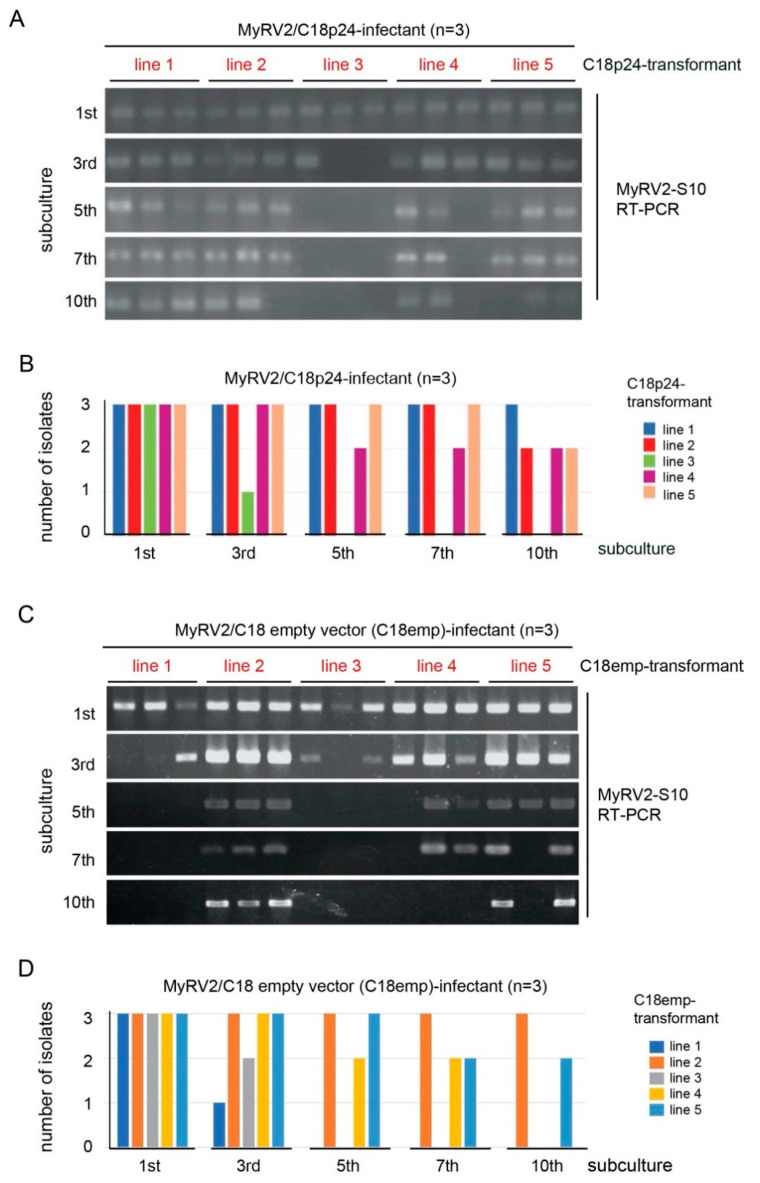
Enhanced infection stability of MyRV2 by transgenic expression of CHV4-C18 p24. (**A**,**C**) Detection of MyRV2 infection by RT-PCR in C18 transformants with the CHV4 p24 coding domain (C18p24) or the empty vector (C18emp). Virus infection in three transfectant subisolates from 5 independent transformants (lines 1 to 5) was monitored until 10th subcultures by colony RT-PCR. RT-PCR was carried out as described by Aulia et al. [28] using the primers CHV4-287_F and CHV4-853_R. Amplified fragments were analyzed in 1.2% agarose gel electrophoresis. (**B**,**D**) MyRV2 Infectivity after successive fungal subculture in C18p24 transformants (**B**) and transformants with the empty vector (C18emp) (**D**).

**Figure 6 biology-10-00100-f006:**
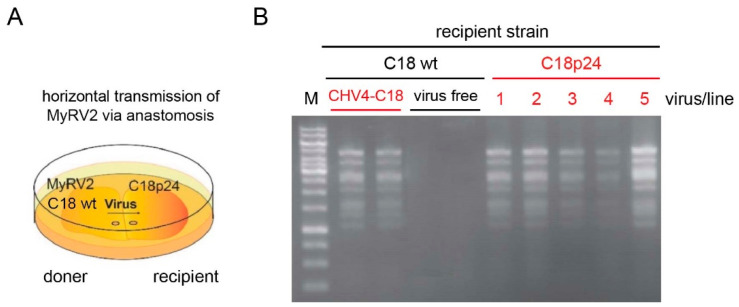
Transgenic expression of CHV4-C18 p24 in recipient *C. parasitica* strains facilitates horizontal transmission of MyRV2 via anastomosis. Non-transgenic donor isolates containing MyRV2 (left side of panel **A**) were co-cultured on PDA with CHV4-C18 p24 transformants (C18p24) as recipients (right side of panel **A**), or with virus-free (C18-VF) and CHV4-C18-infected (C18/CHV4) recipients as controls. Following anastomosis, subcultures were obtained from the recipient side and tested for MyRV2 infection by conventional dsRNA extraction and agarose gel electrophoresis (panel **B**).

**Table 1 biology-10-00100-t001:** Fungal and viral strains used in this study.

Strain	Description	Reference or Source
Fungal		
C18	*Cryphonectria parasitica* field strain doubly infected by MyRV2 and CHV4-C18	[29]
C18-VF	Virus-free single conidial isolate of C18	
C18p24	C18 transformed with the CHV4 p24 coding domain	This study
C18/MyRV2	C18 singly infected by MyRV2	[28]
C18/CHV4	C18 singly infected by CHV4-C18	[28]
C18/MyRV2+CHV4	C18 doubly infected by MyRV2 and CHV4	[28]
C18/Δp69	C18 singly infected by Δp69	This study
C18p24/MyRV2	C18p24 infected by MyRV2	This study
EP155	Standard strain of *Cryphonectria parasitica* (virus-free)	ATCC 38755
C18 Δ*dcl2*	*dcl2* knock-out mutant of C18 (RNA silencing defective, virus-free)	[28]
Viral		
MyRV2	Strain belonging to the species *Mycoreovirus 2* within the genus *Mycoreovirus*	[29]
CHV4-C18	Strain of the species *Cryphonectria hypovirus 4*	[28]
CHV1-EP713	Prototype of the family *Hypoviridae*	[32]
CHV1-∆p69	ORF-A deletion mutant of CHV1-EP713 lacking the p29 and p40 coding domain	[33]

## Data Availability

The data presented in this study in article 10x or supplementary materials as above.

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
