# Peer review of "Identification of an RNA Silencing Suppressor Encoded by a Symptomless Fungal Hypovirus, Cryphonectria Hypovirus 4"

_biology, 2021, doi:10.3390/biology10020100_

Round 1

Reviewer 1 Report

The manuscript titled "Identification of an RNA silencing suppressor encoded by a symptomless

fungal hypovirus, Cryphonectria hypovirus" shows interesting and novel outcomes caused by

symptomless CH4 when coinfecting with an other virus MyRV2 which causes unstable infection.

This coinfection leads to facilitate stable infection and efficient transmission of MyRV2.

Moreover, CHV4 polyprotein called p24 is shown as a suppressor of host antiviral RNA

silencing.

Previously, the review comments were responded well in accordance.

Reviewer 2 Report

The authors have made all reasonable improvements. I think this manuscript should be accepted.

(note: This is a resubmitted manuscript. This reviewer (Academic Editor) gave such suggestion based on the original manuscript, the revised manuscript and authors' reply to all original reviewers' comments.)

This manuscript is a resubmission of an earlier submission. The following is a list of the peer review reports and author responses from that submission.

Round 1

Reviewer 1 Report

The manuscript by Aulia et al. describes the role of RNA suppression by the mycovirus CHV4 and its role in maintaining the RNAi-inducing virus MyRV2 by the action of the putative papain-like cysteine protease p24. The authors show the role of p24 in reducing gene expression of RNAi associated promoters and that p24 can help to support MyRV4 maintenance through disruption of RNAi silencing. In addition, p24 appears to be as effective as CHV4 at promoting MyRV2 horizontal transmission, presumably by anastomosis. Purification of p24 was impressive, but data was lacking to show the purification process that would unambiguously show the purified and cleaved p24. The significance of the changes in small RNAs was not clear from the experiments or the discussion and will need a better explanation. Some experiments lacked adequate quantification of the data and communication of error associated with measurements and some experiments would have benefitted from the inclusion of more control conditions. Overall, the manuscript was reasonably well written, but there are key control experiments that have been omitted that should be included.  I propose the following major and minor edits to improve the quality of the manuscript:

Major comments

Line 178 – This section qualitatively describes a fluorescent reported assay, but only shows representative images. The authors should quantify the mean fluorescent intensity of multiple images for each condition to get a robust measure of GFP induction.

Figure 1C – the figure quantifies the intensity of the Northern blot signal and states “~0”. If the authors are quantifying their data in a rigorous manner, this value should be 0 and used as a background subtraction during the quantification of other lanes. I would like to see the error associated with the quantification of these data. Image data presented can be misleading due to the presence of saturated pixels.

Line 220 – how is the presented data similar to the references article? Also this should really be a discussion point. Comparisons with previously published data are mostly not appropriate for the results section.

Line 223 – no evidence for this statement – speculation that should be moved to the discussion.

Line 231 – the author describes a 2-fold difference as a “modest positive impact” on viral RNAs. The data only counts <4 reads per million reads to support this conclusion (. What is the experimental noise from such an experiment? How can the authors be sure that this is not just biological noise?

Figure 3 – It is not quite clear whether the indicated bands on the crowded Coomassie-stained gel are actually the proteins of interest. As the protein is tagged at each end with epitopes, I suggest that the authors confirm the identity of all constructs using Western blotting to remove ambiguity from the data.

Figure 3 – no data shown for the purification of the CHV4 polyprotein prior to protein sequencing, even though the purification is described in the legend.

Line 293 – The authors describe p24 as a “putative homolog of p29”. Is there any sequence similarity or structural homology that could be reported in the manuscript to support this? Is there any evolutionary relationship?

Figure 4 – Please quantify the bands and include error calculations. Some of the bands in the cell lines expressing p24 look like the CHV4 negative lanes and so I expect a high degree of error. With replicates I would expect the authors to run statistical tests to prove the significance of their findings and that p24 can significantly reduce dcl2 mRNA.

Line 317 and 299 – include the missing data so that the empty vector control is included.

Line 329 – to enable the reader to make meaningful comparisons between datasets, please move Figure S2 data to Figure 5 to enable direct comparison of the stability of the virus with and without p24.

Line 346 – If MyRV2 is “rarely” seen to move laterally, then this is important data to enable the comparison to the frequency of horizontal transfer. These data need to be included in the paper and not cited as “unpublished”.

Figure 6 – the authors should include an empty vector negative control as they have done in previous figures.

Minor comments

Figure 2  - Panel C, bottom two graphs, which two viruses are being used in the co-infection, it is not written in the legend or on the graphs.

Figure 2 – Some places it refers to an infected strain as CHV4/C18 (e.g. Figure 2), other times as CHV4-C18 (e.g. Figure 1), be consistent

Figure 2 – Panel A, CHV1 data – do not shorten the axis at the exact boundary between the + and – small RNA data. Move it down so that the reader can see the divide clearly.

Figure 2 – Small RNA graph x-axis overlap – they should be separated for clarity

Figure 2 – “single infection” not “singles infection”

Figure 2 – Missing a marker for panel (C) in the legend.

Line 203 – an extra hard return that should be deleted

Figure 1 – Does the ‘-‘ dataset in panel A represent a "no virus" control? In other figures, it would be labeled as “no virus”, please be consistent.

Line 187 – plasmids are not “transformed”, organisms are transformed by the introduction of forign DNA, please alter the language accordingly.

In several places in the manuscript, special characters have been deleted. Line 174, 175, 177 are all missing the ‘delta’ symbol. Line 158 is missing “X”

Figure S1 is pixelated – please include a high-resolution version that can be read.

No methods cited or described for dsRNA extraction, please include either or both of them.

Reviewer 2 Report

The manuscript titled "Identification of an RNA silencing suppressor encoded by a symptomless fungal hypovirus, Cryphonectria hypovirus" shows interesting and novel outcomes caused by symptomless CH4 when coinfecting with an other virus MyRV2 which causes unstable infection.
This coinfection leads to facilitate stable infection and efficient transmission of MyRV2. Moreover, CHV4 polyprotein called p24 is shown as a suppressor of host antiviral RNA silencing. 

I have following minor comments,

Line 158: remove 22 mm as mentioned twice. 

Have there been any symptoms or phenotypic alterations as a result of coninfection?
Is it possible to add phenotypic profile after CH4 coinfection with Myr2?
